# Fit for our purpose, not yours: Benchmark for a low-resource, Indigenous language

**Suzanne Duncan**
Te Reo Irirangi o Te Hiku o Te Ika
Kaitāia, NZ 0410
suzanne@tehiku.co.nz

**Gianna Leoni**
Te Reo Irirangi o Te Hiku o te Ika
Kaitāia, NZ 0410
gianna@tehiku.co.nz

**Lee Steven**
Te Reo Irirangi o Te Hiku o te Ika
Kaitāia, NZ 0410
lee@tehiku.co.nz

**Keoni Mahelona**
Te Reo Irirangi o Te Hiku o te Ika
Kaitāia, NZ 0410
keoni@tehiku.co.nz

**Peter-Lucas Jones**
Te Reo Irirangi o Te Hiku o te Ika
Kaitāia, NZ 0410
peterlucas@tehiku.co.nz

## Abstract

Influential and popular benchmarks in AI are largely irrelevant to developing NLP tools for low-resource, Indigenous languages. With the primary goal of measuring the performance of general-purpose AI systems, these benchmarks fail to give due consideration and care to individual language communities, especially low-resource languages. The datasets contain numerous grammatical and orthographic errors, poor pronunciation, limited vocabulary, and the content lacks cultural relevance to the language community. To overcome the issues with these benchmarks, we have created a dataset for the Māori language (the Indigenous language of Aotearoa/New Zealand) to pursue NLP tools that are 'fit-for-our-purpose'. This paper demonstrates how low-resourced, Indigenous languages can develop tailored, high-quality benchmarks that; i. Reflect the unique characteristics of their language, ii. Reflect the diversity of speakers in the language community, iii. Support the aspirations for the tools they are developing and their language revitalisation efforts. All of which sit within a broader understanding of the impact of colonisation on their language.

## 1   Introduction

Benchmarking is used in natural language processing (NLP) and machine learning (ML) as an objective lens to measure or track the progress and accuracy of a model. However, standardised open-source datasets do not always provide appropriate forms of benchmarking as they rely on large datasets and favour a small group of high-resource languages, like English (Mdhaffar et al., 2024). This means that they fall short on low-resource, under-resourced, minority, colonised and Indigenous languages. It has been generally agreed that the development of NLP tools requires large, annotated datasets. Yet, scarcity of data is a common issue in the endeavour to create quality tools because NLP research concentrates on 20 of the more widely spoken languages of the more than 7000 languages spoken throughout the world (Magueresse et al., 2020; Mdhaffar et al., 2024; Joshi et al., 2019; Joshi et al., 2020; Bird, 2022; Markl et al., 2024; Nicholas & Bhatia, 2023). This has culminated in a

limited amount of research conducted on low-resource languages, referred to as such because of the lack of data available (Doumboya et al., 2021). For low-resource languages, and in particular Indigenous languages, the valorised benchmarks are often laden with linguistic errors, perpetuate poor pronunciation, and have limited vocabulary because the content lacks cultural and linguistic intelligence.

This paper provides an overview of benchmarking for both high- and low-resource languages. It highlights how 'cutting-edge', popular benchmarks are fit for some peoples' purposes, but not fit for ours. In particular, we discuss FLEURS, the Few-shot Learning Evaluation of Universal Representations of Speech benchmark, and the issues we encountered, using the Māori language (the Indigenous language of Aotearoa/New Zealand) as a case study. The paper then introduces Te Reo Irirangi o Te Hiku o te Ika's specifically curated benchmark, the first of its kind for the Māori language. The high-quality benchmark was created to address the resource gap that is fit-for-OUR-purpose and is culturally and ethically more appropriate for the Māori language community. It is based on our political history, has thorough quality assurance and reporting and a methodology that includes conscious decision-making to ensure high-quality outcomes. This paper aims to provide a clear example for other language communities in similar positions to replicate and curate their own benchmarks. It provides an example of how quality over quantity in curating a tailored benchmark is more beneficial than measuring models and tools against more commonly used benchmarks.

## 2   Part one: Fit for their purpose

Benchmarking is crucial to ensuring progress in ML and NLP and it is used as a reference point to compare and evaluate performance against one or multiple metrics. Some benchmarks contain multiple datasets, that "operate as stand-ins for a range of anointed common problems that are frequently framed as foundational milestones on the path towards flexible and generalizable AI systems" (Raji et al., 2021: 1). The true effectiveness of a benchmark is measured by how it helps us understand how systems work and don't work. However, the overly generalised nature of many of the established benchmarks inhibits any recognition of the unique cultural circumstances of individual language communities and their speakers (Nicholas, 2024). Many of these influential benchmarks have been put on a pedestal as a target that the entire field should be striving for state-of-the-art performance on. High-resource languages are capable of achieving human-level proficiency on numerous NLP tasks on the popular benchmarks because they have the privilege of significant amounts of data and infrastructure. Many of those working with low-resource languages can get distracted by this goal and become occupied with chasing good metrics rather than respecting the aims and desires of the language community (Thomas & Uminsky, 2020; Raji et al., 2021; Leoni et al., 2024).

Data quantity is still perpetuated as crucial to the progress and success of datasets, large language models (LLM) and systems. This has led to extensive harvesting and scraping of the internet to gather 'enough' data for different languages (Wiechetek et al., 2024). This 'crawling' of the internet is an inexpensive task for high-resource languages that have millions of websites written in their languages (Bustamante et al., 2020). English-language texts were mass-produced throughout the 19th and 20th centuries, leading to English being 'the language of the internet' (Nicholas & Bhatia, 2023). Truly low-resource languages have very few speakers and little to no written text (and minimal or none in digital forms) (Bird, 2022; Bustamante et al., 2020; Wiechetek et al., 2024). Many of these language communities only established writing systems upon contact with their colonisers. Whilst some have established a form of standardisation of their languages, many have not and most of the early written texts available, that offer less-diluted versions of their languages, were written by non-native speakers.

This has led to a growing 'saviour' complex in speech and language technology circles in the search for language equality (Markl et al., 2024). The concepts of 'no language left behind' and 'language technology for all' feed the egos of language technologists who are looking to be the first to solve machine translation and build speech recognition tools for all languages. Often the end goal of building valuable and useful systems and tools is forgotten in the bid to achieve incremental improvements in performance on benchmarks (Joshi et al., 2019). Most models and tools fail to recognise the vast and unique difference between language communities and their historical, cultural, political and sociolinguistic contexts (Bird, 2022). When technology is designed for a different social, cultural and linguistic context, it leads to language modelling bias and technology full of linguistic

and cultural errors (Markl et al., 2024). The needs, desires and overall linguistic goals of a language community must be at the forefront of decision-making to ensure the development of high-quality and useful language tools and enable rather than inhibit language communities to receive information and communicate in their preferred languages (Joshi et al., 2019). Examples of this include First Languages AI Reality (FLAIR) (Mila, 2024) and Wiechetek et al. (2024).

## 2.1 FLoRes and FLEURS

To avoid leaving any language behind, multilingual models have become a focus for language technologists, which has resulted in the creation of multilingual datasets, many of which are machine-translated. These multilingual models are predominantly trained on English-language text, "which leads them to apply an Anglocentric lens onto their analysis of texts from non-English linguistic and cultural contexts" (Nicholas & Bhatia, 2023: 1). Benchmarking against machine-translated benchmarks only demonstrates how well a model measures against a machine-translated text, rather than the real language that speakers from a language community use. This issue is of particular relevance to low-resource languages because machine translations are usually poor and inaccurate (Nicholas, 2024) and overlook the unique cultural and linguistic contexts of those languages (Kim et al., 2024).

The FLoRes-101 evaluation benchmark contains 3001 written sentences that were extracted from Wikipedia and translated into 101 languages by professional translators. According to the creators the dataset "enables better assessment of model quality on the long tail of low-resource languages, including the evaluation of many-to-many multilingual translation systems, as all translations are fully aligned" (Goyal et al., 2022: 522). The FLEURS speech dataset uses 2009 sentences from the FLoRes-101 benchmark.

We qualitatively assessed the datasets in the same way we would assess the writing and proficiency level of a Māori language student.

### 2.1.1 Māori worldview

The content used in the FLoRes-101 and FLEURS datasets comes from a range of Wikipedia pages. Because the data used comes from Wikipedia and is translated, the perspective and, therefore, language do not reflect a Māori worldview. There are only four instances where Māori people are mentioned (see Figure 2) and the English version is missing the macrons on Māori. All four examples refer to the arrival of Māori to New Zealand, a historical event that happened around 800 years ago. There is no 'content' that discusses Māori life or Māori realities. The reporting nature of the content also means that the language is very formal, it doesn't reflect a conversational, everyday use of the Māori language or language used in Māori settings.

### 2.1.2 Vocabulary range

Because the content comes from Wikipedia, it is mostly biographical or reports of events and therefore the vocabulary used is largely limited to that domain. The competency of the translator is also apparent when considering the vocabulary used, for example, when describing heavy traffic the translator has used the word 'taumaha'. Taumaha describes something that is physically or emotionally heavy. There are other Māori words for congestion which are more appropriate for this context, such as opuru or kūkā.

### 2.1.3 Common mistakes

The translation included several grammatical errors which are common for beginner to intermediate language learners. These types of mistakes usually occur when trying to create a literal translation and following an English grammar structure for the sentence.

For example, the particles 'i', 'ki' and 'mō' have been used incorrectly in the following sentence:

**Māori translation from dataset:** 'E āhei ana ngā tauira ki te mahi ki tāna i pai ai, ki tāna i kī ai mō te āhua ki ngā pārongo tohutohu'

**English equivalent from the dataset:** 'It allows students the ability to work at their own pace and control the pace of instructional information.'

There are also a large number of incorrect applications of macrons, here are a few examples with the correct spelling in brackets: Ēngari (engari), Tōnā (tōna), He tīno akiaki ana (tino), Tāunga (taunga)

### 2.1.4 Grammar structures

Throughout the datasets, we see a limited range of grammatical structures and many follow the grammar structure of an English sentence. There are several examples where the translated Māori sentence would be at a 'beginner' level. One of these is the saying "**Mai** i tērā wā" (from that time); this follows an incorrect adoption of the word 'mai' for the word 'from', the sentence could drop the word 'mai' and would be more correct.

### 2.1.5 Pronunciation

There appear to only be four speakers in the data set. Two speakers have a good degree of pronunciation and two speakers have poor pronunciation. There are also instances where the speaker does not correctly read the target sentence and therefore creates grammatically incorrect data. For example:

**FLEURS Target Sentence:** **I** ētahi wā **ka** tino **hiahia** te kaiako **kia** haere ōna ākonga ki tōna taha...

**What the speaker says:** **E** ētahi wā **kia** tino **haehae kē** kaiako **ki** haere ōna ākonga ki tōna taha...

### 2.2 Summary

These datasets reiterate incorrect use of grammar, vocabulary, spelling, macrons and pronunciation which harms the quality of the Māori language because any group can access and reproduce these sentences. The authors have claimed to create a "high quality" (Goyal et al., 2022: 522) dataset for a multitude of languages. If this were true, it would provide an option to train and evaluate models on high-quality examples of speech in low-resource languages (Nicholas and Bhatia, 2023). However, this analysis demonstrates that these are an extremely poor pair of benchmarks for the Māori language. As a language community, if we want to create high-quality tools with high standards we cannot measure this technology against generalised benchmarks. When done correctly, benchmarking allows us to survey a landscape, therefore "the more we can re-frame, contextualize and appropriately scope these datasets, the more useful they will become as an informative dimension to more impactful algorithmic development and alternative evaluation methods" (Raji et al., 2021: 10). Comparisons between FLEURS and our own benchmark will be made in the following section.

## 3    Part two: Fit for OUR purpose

To overcome the inherent shortcomings of popular benchmarks, we have created our own language-centric benchmark as a reliable and fair method of evaluation and performance reporting for the Māori language. This novel benchmark has been used to track the progress of the Māori language, and now bilingual (with English) ASR model.

For many low-resource languages, and in particular Indigenous languages like Māori, there is no separating language from culture, identity and history. There were consistent and intentional attempts to eradicate the Māori language in favour of English, beginning in the late 19th century, heightened in the 20th century, and continuing to this day (Higgins et al. 2014; Keenan 2012; Winitana 2011; Walker 1990; Te Rito 2008; Leoni, 2016). Since the 1970s, Māori have made significant attempts to revitalise the language. As of 2024, there are approximately 71,000 highly proficient/fluent speakers of the Māori language (1.5% of New Zealand's total population), and 190,000 people who can hold a conversation (7% of the population). These numbers represent a growing population of people who want to engage with technology in the Māori language. Despite these efforts, Māori is still very much a low-resource language. There is limited data, limited resources, and limited natural language pre-processing (NLPre) tools (Te Reo Irirangi o Te Hiku o te Ika, 2022). This paper has discussed the expectation of having large quantities of data for ML and NLP; this amount of data is non-existent in the Māori language.

Te Reo Irirangi o Te Hiku o te Ika (Te Hiku Media) is a tribal radio station based in Kaitāia, New Zealand. We have been tasked by our elders to create ethically sound and culturally appropriate tools for our people (Te Reo Irirangi o Te Hiku o te Ika, 2022). There has been a commitment to revitalising the language through the tribal radio station since the 1990s. The creation of a segment called 'Te

Hāora o te Reo Māori' (the Māori language hour) included airing live and recorded interviews with native-speaking elders. Despite the topics of conversation being intriguing and significant to the listeners, the main purpose of the show was to share the language spoken by the elders in the hopes of revitalising the language of the region. These and many other recordings have been kept and digitised by the organisation. They, along with the contemporary radio content captured daily, now form the largest archive in tribal radio network (Jones et al., 2023b).

## 3.1 Māori ASR development

Despite advances in multilingual ASR models, the principles of data sovereignty compel us to use data collected with proper permissions. It became apparent that we needed to create our own Māori language speech technology, including an ASR model to share our archive and make it more accessible (see Te Reo Irirangi o Te Hiku o te Ika, 2022). And so, we created the first iteration of a speech-to-text (STT) model for the Māori language. It was initially built using Mozilla's Deepspeech architecture, which relied on recurrent neural networks (RNN). The model was trained on nearly 400 hours of labelled data gathered during a community-led crowd-sourcing campaign.

In order to create a tool that reflected our language community, a decision was then made to create a bilingual (Māori and NZ English) STT model. We trained another model from scratch using our own data due to the limited availability of external labelled Māori data. We reasoned that the model would still be competitive when trained with bilingual and code-switched data in just two languages because our data is high-quality. The current STT iteration is now bilingual. We chose the fastformer variant of the conformer-rnnt-large model from the NVIDIA NeMo framework, a convolution-augmented transformer (Gulati et al., 2020). Further decisions involved selecting a decoder and a bilingual BPE tokenizer with a 1024 vocabulary size, along with an adaptive length synchronous decoding (ALSD) beam search for optimal performance.

Table 1: Number of parameters in multi lingual models

| Model | Parameter Count (millions) |
|---|---|
| Whisper-large-v2 | 1,550 |
| Whisper-large-v3 | 1,550 |
| MMS | 1,000 |
| Conformer-Large | 120 |

## 3.2 Data creation and labelling

There were many decisions made during the model creation process to ensure that we truly contribute to the restoration of the Māori language. This included that the newer models needed to exemplify a native speaker sound; a type of language and a prosody that would be viewed as 'aspirational' for second language learners (Jones et al., 2023a; Leoni et al., 2024). This required us to re-think how we collated and curated data in order to gather enough data to train, validate and test the model. It initiated a concentrated effort to create data through human transcription and labelling Māori language audio from our archive.

When the human transcription process began, the team had not decided on a consistent orthography for transcriptions. This prompted the creation of a set of guidelines by language experts who work in Te Hiku Media's data science team to develop consistency. The guidelines include instructions on orthography, how to write numbers, an explanation of verbatim transcriptions, and what resources to use to decipher unknown phrases or words. The guidelines also outlined what was considered an "utterance" and how the data would be segmented into utterances. This process initiated conversations on quality assurance and a peer-review process being established. All transcripts would be transcribed by one person, and reviewed by one of the senior language experts. When the audio was tough to decipher, the pair would discuss the possible options and agree on the final label using knowledge of the language and content. This process has ensured that the data curated is high quality and therefore positively impacts the quality of the model.

### 3.3   Te Taumata - The Benchmark

Through this process, it became apparent that the creation of a high-quality dataset to benchmark was necessary to evaluate the ASR model's performance. Broadly, the benchmark dataset needed to represent a range of attributes, but most importantly, the benchmark needed to measure how well the ASR worked on the type of language that is important to the language community requiring the identification and development of a series of sub-benchmarks with key characteristics, while also ensuring balanced gender and age representation.

The first and most important sub-benchmark of the dataset was for Native Sound and included four native Māori language speakers. These speakers were born before 1930 and they represent the native sound of a generation who primarily learnt their language without the influence of English in their homes. Speakers were also selected from different parts of the local linguistic ecosystem. Vehicular versus vernacular (Fishman, 2001; Bird, 2022) representation ensured that there were speakers who had participated in the public sphere of education, broadcasting and politics as well as those who spoke more informally and embodied the language of the home.

Three male and three female speakers (two were below 30 years of age, and four were between 31 and 65 years of age) were selected from our collection to reflect a new generation of speakers. Of the growing number of Māori language speakers, second language learners and contemporary first language speakers are the largest groupings, and those most likely to engage in this type of technology (Te Reo Irirangi o Te Hiku o te Ika, 2022). Second language learners are usually speakers who have learnt the language later in life, usually through one of the main university or community-based programs. First language speakers represent the group of largely under 40 year-olds that learnt the language as a child through one of the language revitalisation initiatives such as kohanga reo (Māori immersion early childhood education) and/or kura kaupapa (Māori immersion primary and secondary education). Both groups of speakers have unique prosody and characteristics important to capture and benchmark.

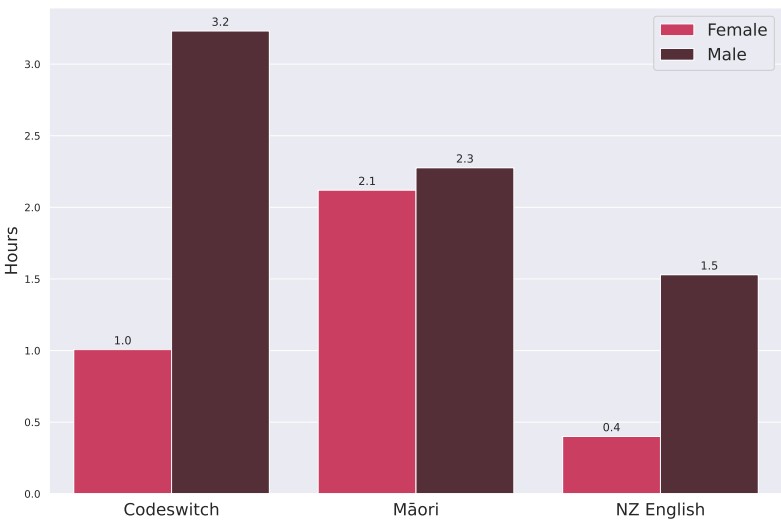

Figure 1: Number of hours per gender by language

Codeswitching is a prominent feature of Māori language speaker communication. When deciding to include a bilingual sub-benchmark, the initial idea was to first include New Zealand English as a standalone sub-benchmark. Language technology is better suited to high-status varieties of high-resource language speakers (Markl et al., 2024). This is problematic for New Zealand because the speech technology trained in English is not trained in the New Zealand English accent. Speakers were chosen similarly to the native and second language speakers, but first and foremost, there was a conscious decision to include what we determine as 'true Kiwi accents'. We selected four female and four male voices to represent this sub-benchmark. Because we had limited data of this kind in our archive, age was not considered, however all sit within the 31 to 65 age range.

The next step involved incorporating bilingual data into the corpus. This was a significant challenge for the team, as there is very limited truly bilingual codeswitching data. Upon searching for potential corpus, it became clear that truly bilingual codeswitching only occurs in speech. The voices that represent this benchmark are primarily male, because the best source was from meetings and gatherings held in the community. This is where you could hear the natural occurrence of consistent codeswitching and in the Māori world, these roles are primarily held by males. There are 10 male voices and 3 female voices, all aged older than 31. This data is significantly different from the interviews which form the majority of the previous data but bridged the bilingual sub-benchmark gap in the dataset which was the priority.

The content of the dataset is wide-ranging, the native speakers talk about their lives and upbringings, forgotten plants used to aid pain and what it was like to go overseas and to war. The contemporary speakers discuss politics, the entertainment industry, healthcare and education. There are also examples of speech from meetings and gatherings held in the community. These were recorded instances of Māori cultural contexts that reflect an important aspect of the language community.

The performance of the ASR is analysed both qualitatively and quantitatively. The word error rate (WER) results from a report are used to compare the initial benchmarked results from the existing model to the proposed model/s. The report is presented with target sentences from the benchmark, and whether the previous and proposed models perform better, the same or worse. The benchmark is also measured against the various Whisper models from Open AI. This provides us with WER metrics to ascertain how accurate our ASR models are compared to the Whisper models and track our progress. The performance of the ASR model has demonstrated significant progress against the benchmark (Jones et al., 2023a). Since the first iteration, there has been a substantial drop in the WER of the ASR model from 27% to 10%.

## 4    Discussion

The creation and curation of Te Taumata relied heavily on the knowledge of the Māori language data specialists. We are familiar with the genealogy of the people within the archive and with the archive in general and have a thorough understanding of the wider sociolinguistic environment in Aotearoa/New Zealand. This has resulted in a benchmark that i. Reflects the unique characteristics of our language ii. Reflects the diversity of speakers in the language community, iii. Support the aspirations for the tools we are developing and our language revitalisation efforts. This ultimately leads to NLP tools that are fit for the language community.

This paper has discussed the issues of influential benchmarks and how they are not fit for purpose for low-resource languages, using a critique of FLoRes and FLEURS as an example. It then introduced Te Taumata, a benchmark established for Māori language ASR evaluation. The following section brings these sections together to demonstrate how the creation of language-specific benchmarks is more relevant and culturally appropriate than valorised benchmarks with two distinct features of the benchmark in comparison to FLEURS.

### 4.1    Unique characteristics

The guidelines developed for the curation and labelling of all of the data and subsequently, the benchmarking dataset, included making decisions on the length and characteristics of the 'utterances'. It was important for the segmentation of utterances to reflect natural speech and consider the way this had changed over time. This particular dataset consists of primarily interviews and conversational language where the speakers talk over each other. There is a significant portion of utterances that are single or two-word responses or exchanges between the speakers (e.g. 'āe' (yes) when agreeing with a question), this can be seen in Figure 2. below.

Native speakers, in particular, in the dataset speak slowly, and whilst ASR models might believe that taking a break or breathing indicates the end of the sentence, the human transcribers and reviewers were meticulous in ensuring that whole thoughts or ideas were completed before splitting an utterance. Contemporary speakers tended to roll sentences together with very short or indistinct breaks in between. This made segmentation difficult but attempts were made to reflect this unique characteristic in the way utterances were made. This has resulted in a unique 'shape' of the Māori language utterances in the dataset.

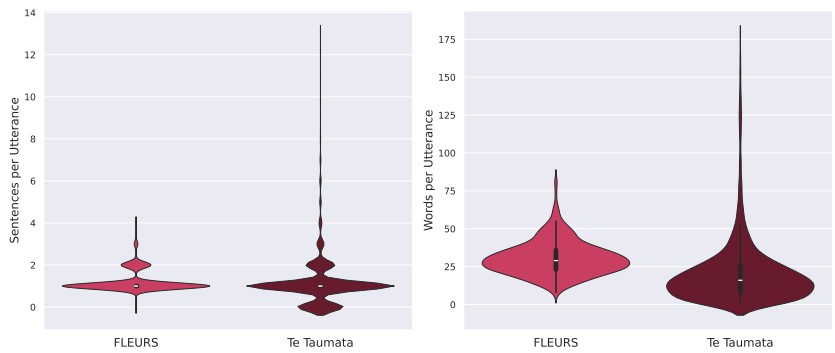

Figure 2: Overall comparison of sentence and word distributions per utterances.

Table 2: Data comparison of Te Taumata vs FLEURS

|  | Te Taumata | FLEURS |
|---|---|---|
| No. of sentences | 4835 | 1211 |
| No. of unique sentences | 4588 | 421 |
| No. of words | 89,277 | 30,794 |
| No. of unique words | 6,402 | 2,033 |
| No. of speakers | 37 (23 male, 14 female) | 4 |

When the same analysis is conducted on the FLEURS benchmark dataset, the 'shape' of the utterances is starkly different from that of Te Taumata. FLEURS utterances are longer with more words. This is reflective of the type of data not being spoken language one would experience in real-life contexts.

Doing it this way is reflective of the actual language used every day by the language community and creates tools that are wanted by the language community. Throughout developing these tools, the language community has asked for applications such as captioning, and transcription of audio archives. Ensuring the benchmark reflects natural language supports decision-making towards developing tools relevant to the community.

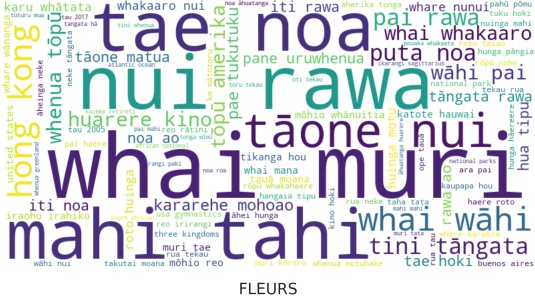

FLEURS

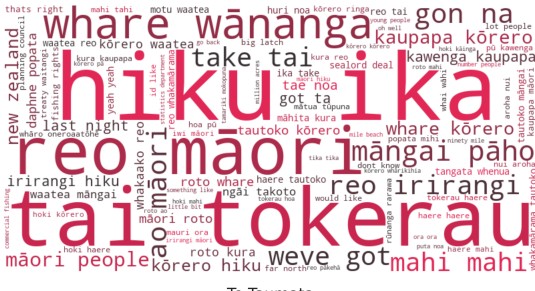

Te Taumata

Figure 3: Word cloud of the top 100 bigrams

A comparison of the top 100 bigrams shows how different the language within the two benchmarks is. The top words in FLEURS are a strange combination of modifiers and conjunctions, as well as, international placenames and topics. The top words in the Te Taumata benchmarks relate to the source of the data, such as placenames from the far north of New Zealand and words that relate to the tribal radio network.

## 4.2 Reflects the diversity of speakers

Like many colonised languages, the Māori language experienced a contraction and is now undergoing a 're-expansion'. As government policies made a significant impact on the number of speakers, we also saw a homogenisation of the language take place. Furthermore, the language was largely only spoken at a high proficiency amongst older speakers. With revitalisation efforts, the language is experiencing a re-expansion, where tribal or regional diversity is being reinvigorated, words and phrases that had long been out of use are revived, all ages are engaging with language and a spectrum of proficiency can now be identified. These are now represented in the sub-benchmarks.

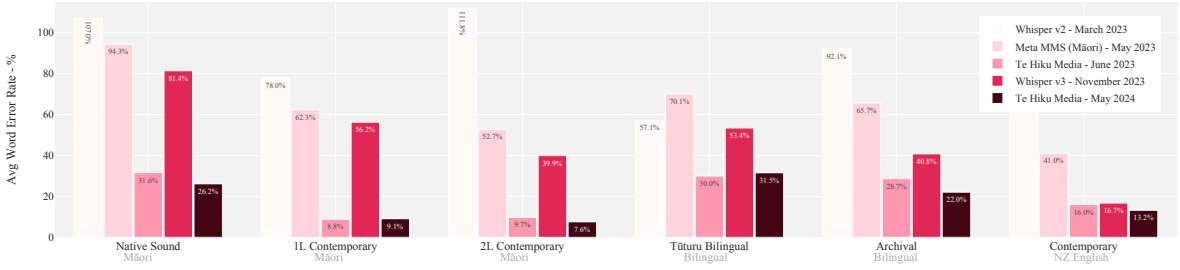

Figure 4: ASR Model performance on Māori Language Sub-Benchmarks

## 4.3 Native speaker aspirations

Understanding performance against the overall Te Taumata benchmark and in comparison to other models, such as Whisper from OpenAI are helpful metrics that can inform high-level strategic decision-making. However, for low-resource languages, the performance and progress against sub-benchmarks can take a higher level of importance when considering the aspirations of a language community trying to save their language.

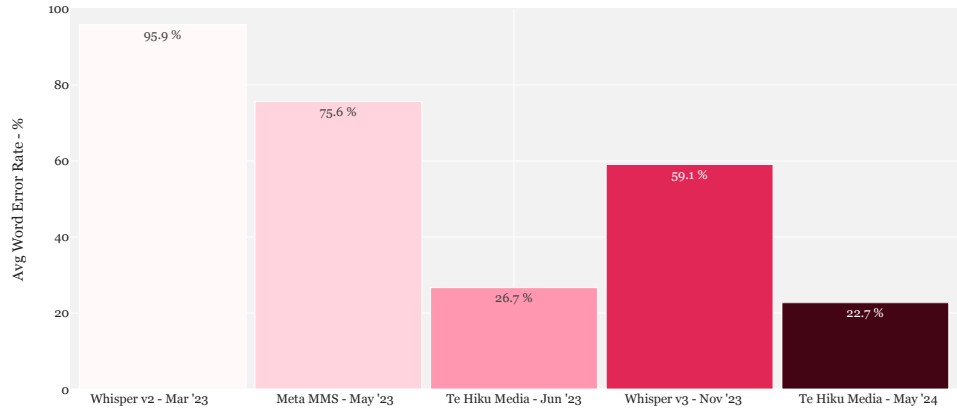

Figure 5: ASR Model performance on Māori Language Benchmark

Restoring the native speaker sound was identified as the main aim for the development of a Māori ASR tool (Jones et al., 2023a; Leoni et al., 2024). It was vital therefore that there was a Native Sound sub-benchmark to continually measure our progress towards that purpose. It was also important to see performance across all of the sub-benchmarks so we could make informed decisions about the next steps for the ASR development and data collection.

As we can see from Figure 5, model performance on the Native Sound sub-benchmark has been steadily improving. While an improved Native Sound WER has seen a slight decline in progress against the First Language Contemporary and Bilingual benchmarks, with the Native Sound being a priority for the language community, such a loss in performance is considered acceptable.

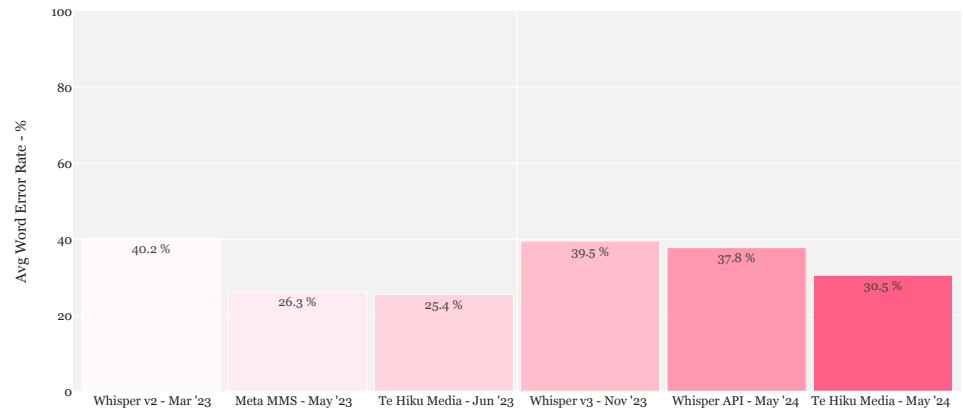

Figure 6: ASR Model performance on FLEURS

Interestingly, as the Māori ASR model began to improve against the Native Sound sub-benchmark it also began to get progressively worse against the FLEURS benchmark (see Figure 6). This could be explained by the lack of Native Sound data within the FLEURS dataset, further demonstrating how not-fit-for-purpose the benchmark is.

## 4.4  Limitations

Low-resource languages are such because of the limited data available for training models, let alone establishing a quality benchmark.To achieve a high quality benchmark dataset that takes into careful consideration, there is significantly more effort required than utilising existing benchmarks that do not. This benchmark was curated to represent the diversity of the community as much as possible. This includes getting representation of various age groups, gender representation, levels of fluency and domains of language including formal and informal language. This may not always be possible for other low-resource languages, however, it should be noted that new sub-benchmarks can and should be added as new data is made available.

The current benchmark does have a bias towards the regional variation of the Māori language found in the far north of the country. Therefore, future work for the benchmark will be to expand the dialectal diversity. Further to this, the NZ English and bilingual benchmark has a larger male representation. This is in part due to the challenges of accessing data. Our collection has always focused on the Māori language, and there was limited quality NZ English options. The bilingual data we have used for this sub-benchmark is from events where men were the dominant speakers. Culturally this often occurs at events that are held on marae and larger gatherings.

## 5  Conclusion

To address the issue of poor-quality, generalised benchmarks, we specifically designed a niche benchmark as a fair and reliable method of evaluating the performance of Māori language ASR tools that are fit for a Māori purpose. This paper demonstrates how low-resource languages can and should create benchmarking datasets to measure their models and tools based on their purpose. We hope that this paper provides a clear template of the type of decisions that need to be made throughout the process to ensure a high-quality benchmark is created, that is language and community-centric.

## Acknowledgements

We acknowledge the continuing support from the five tribes, the trustees that represent them and the many community members of Te Hiku o te Ika who contribute to this work.

This work was funded by the New Zealand Ministry for Business, Innovation and Employment through the Strategic Science Investment Fund.

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
