# OpenReview forum: "Fit for our purpose, not yours: Benchmark for a low-resource, Indigenous language"
_NeurIPS.cc/2024/Datasets_and_Benchmarks_Track — NeurIPS 2024 Track Datasets and Benchmarks Poster_

### Official Review · Reviewer_bjZD · 2024-07-19
**A benchmark designed by and for an Indigenous language community**

**Rating:** 10
**Confidence:** 5

**Review:**

Pros:
Quality:  This is a thoughtful and insightful contribution to the growing literature on Indigenous language-community-led language technology development which offers useful transparency on the development process from initial commissioning ('[Te Hiku Media radio station] have been tasked by their elders to create ethically sound and culturally appropriate tools for their local people' (the 'what') through to the establishment of purpose (the 'why') (to support community access to and use of the radio station's language archive for language restoration) and data collection/selection, transcription and quality assurance processes (the 'how').
Clarity: The paper is clearly written. The rationale is clear for each step undertaken, from the establishment of the need for a benchmark of this kind, to the process of curating the benchmark itself.
Originality: The benchmark presented is the first of its kind for the Māori language. The process of cultural consultation itself is original, showing evidence of an agile development process that was highly responsive to the realities of language use in the community. For example, the authors note that the original idea of a New Zealand English sub-benchmark dataset was rejected in favour of a dataset that accurately reflects the code-switching that occurs in spoken Māori.
Significance: There are few Indigenous and low-resource language benchmarks developed by communities for their own purposes and designed to preserve language data sovereignty (the Masakhane initiative for African languages ('grassroots NLP community for Africa, by Africans') is one of the rare exceptions). This benchmark is a significant contribution to an important alternative, not-for-profit stream of NLP technology development.

Cons:
Clarity: There are a couple of opportunities for greater clarity in relation to data preparation and demographics, outlined below.
Significance/Originality: A minor comment. This paper understandably focuses on the context and significance of the development of this benchmark for the Māori language. This means there is less focus on the efforts of other Indigenous language communities in this direction. Given that the paper is intended in part to provide a model for the development of similar benchmarks by other language communities, adding more references to strengthen the acknowledgment of other Indigenous language technology development efforts could support this aim. However, it would be important not to dilute the focus on the cultural and linguistic specificities of the Māori language community, which are one of the strengths of this paper.

**Strengths:**

The specificity of the linguistic and cultural insights provided into the Māori language and language community is one of the most important strengths of this paper. Demonstrating and supporting the breadth and depth of linguistic and cultural diversity that exists in everyday language use provides a corrective to the tendency for (commercial) language technologies to support and promote the most dominant, standard or translatable language structures, meanings, and contexts of use.

The relevance to the broader research community is that research such as this highlights important ethical and social dimensions to language technology development, such as:
1) the ethical dimension of language data sovereignty, which is too often overlooked in the rush to 'include' all languages in commercial language technology, which in some cases may be occurring against the wishes of Indigenous and non-Indigenous, low-resource language community members, raising the question of who benefits from such inclusion, if not the language community;
2) the reality of intra-and inter-language variation and dynamics, such as lexical differences between older and younger populations and bilingual and multilingual code-switching, which are common phenomena across the world's languages, but rarely directly addressed by language technologies or evaluation benchmarks;
3) the importance of what the authors here refer to as 'Native Sound', meaning, model pronunciations of the language as produced by community elders and language teachers, the preservation of which has been identified as 'a priority for the language community'. The authors note that there is a 'lack of Native Sound data within the FLEURS dataset';
4) the importance of training data content that represents the cultural worldview and not only the lexical and grammatical structures of the target language (the authors observe that the Wikipedia data used as the basis of the Flores dataset does not include material relevant to the Māori worldview).

**Additional Feedback:**

No additional feedback.

**Clarity:**

Yes, the paper is clearly expressed (but see the note above about improvements to the clarity of diagrams).

**Correctness:**

The benchmark's data collection and preparation pipeline is adequately described and correctly performed. More details on the ASR architecture could better support replicability by other language communities, but it is not the focus of this paper. Standard evaluation methods are applied (WER).

**Documentation:**

The paper provides sufficient detail on the curation and preparation of the dataset and gold standard transcriptions which form the basis of the Te Taumata benchmark for evaluating ASR performance on a diverse range of Māori language production across older and younger speakers and including bilingual production with code-switching. The paper does mention the necessity of establishing orthographic guidelines and standards to ensure consistent spelling in the benchmark transcriptions, without detailing the complexities or challenges encountered in this process. A future paper might provide this detail, which would undoubtedly be useful for other language communities facing the same challenge of establishing a consensual written standard for the purpose of ASR benchmark creation.

**Ethics:**

I have no ethical concerns to flag.

**Limitations:**

No potential negative societal impacts are apparent and potential positive societal impacts are addressed well. The authors do not explicitly address any limitations on the current and future development of the benchmark and they may wish to consider adding one or more sentences to briefly address these. Efforts to achieve a better gender balance may be one of these (including any plans in relation to the representation of non-binary genders).

**Opportunities For Improvement:**

There are some minor opportunities for improvement.
1. In Figures 1 and 2 comparing word and sentence distributions per utterance in the FLEURS and Te Taumata datasets, the scale on the horizontal axes differ, making the reader's task of comparison harder;
2. More demographic detail is needed. The authors note that "the benchmark dataset needed to represent a range of attributes" but it's unclear which attributes are meant. They observe the importance of "ensuring balanced gender and age representation" but age representation in Te Taumata is not reported in detail (we are told 4 speakers were born before 1930), while gender representation is actually reported as unbalanced (23 male, 14 female, Table 1). No explanation is offered as to why there is this imbalance (the intersectional demographics of gender x age would also be good to report).
3. In section 4.1, the description of Māori utterance splitting for data processing ('human transcribers and reviewers were meticulous in ensuring that whole thoughts or ideas were completed before splitting an utterance') seems to conflict with Figure 1 ('Distribution of sentence counts per utterance') which appears to indicate that between 500 and 1000 utterances had less than one sentence per utterance. If correct, this bears explanation.
4. The current text size and colouring of Figures 3 through 5 impacts readability.
5. Question 6 'Experimental Setting/Details' - the correct answer would appear to be N/A rather than 'Yes'. Same for Q.8.

**Relation To Prior Work:**

This is sufficiently discussed, given the main focus of the paper is on the differences in language quality (as measured by breadth of vocabulary, grammatical structures, representation of worldview and grammatical and pronunciation errors) found in a commercial Māori dataset and the language quality (accuracy, diversity and representativeness) achieved by the Te Taumata Māori benchmark. The benchmark development is broadly contextualised by reference to research on NLP development in minoritized, colonised and Indigenous languages, and commercial efforts aimed at achieving 'language technology for all'.

**Summary And Contributions:**

This paper describes an often-overlooked but essential criterion for an Indigenous language technology benchmark: that, as a benchmark for evaluating technologies for use by the language community in question, it should appropriately and adequately address the language community's needs and aspirations. The authors develop a Māori community-led dataset, Te Taumata, as a resource for Māori language technology development and an exemplar for other minoritized language communities seeking to develop their own language technology. While this is pioneering work that few Indigenous language communities have the resources to undertake (if they wished to), it provides an important articulation of the issues at stake for such communities and the necessity for minoritized language communities to develop and own their language benchmarks.

This contribution argues convincingly that the only way for this to happen is with the deep involvement and leadership of the language community. The authors compare an existing Māori language dataset, developed originally in text for multilingual translation applications (Flores-101) and its extension into speech (FLEURS) using a subset of 67% of sentences from Flores-101 as the texts for recording. They assess both datasets on dimensions including vocabulary coverage, grammatical structures, the worldview represented in the sentences and its relationship to a Māori worldview, and pronunciation. Having demonstrated the shortcomings of the existing datasets, the authors describe the development of a Māori benchmark to track the performance of Māori and bilingual Māori-English ASR systems over time.

---

> ### Author Rebuttal · Authors · 2024-08-16
>
> E mihi atu ana ki a koe i te arotake nei. Thank you immensely for this review. We are grateful for the thoroughness you took in providing suggestions and discussing its strengths; we found it very encouraging. Several of the issues you raised have been included in the global response and are left out of here as they were picked up by some or all reviewers. This rebuttal contains the remaining issues you raised:
>
> We agree that it is important to recognise the efforts of other Indigenous communities. Following line 96, we have added a sentence and references to acknowledge the efforts of groups such as the First Languages AI Reality (FLAIR) Initiative and individuals such as Linda Wiechetek.
>
> Regarding Figures 1 and 2, this was a challenge to visually represent the data as the distributions were so disproportionate. A new figure has been created (now just Figure 1) and is included in the attached PDF.
>
> Figure 1 has been included that shows female and male distributions for each sub-dataset (Māori, English & bilingual). In lines 280-285 we highlight the challenge of finding truly bilingual/codeswitching data. Therefore, we were not able to prioritise ensuring a gender balance in this sub-dataset. For the domains that the bilingual data comes from (formal meetings etc) it is more common for males to speak than female. We have changed the language throughout to explain the imbalance.
>
> The dataset consists primarily of interviews and quite conversational language, therefore there is a significant portion of the utterances that are single or two word responses/exchanges between the speakers, such as, ‘āe’ (yes) or ‘kāo’ (no). Furthermore, they can at times interrupt each other when they are speaking which can disrupt a sentence. We have included this explanation in the paper and the new Figure 1 better demonstrates what we mean.
>
> More details have been added to section 3.1 regarding the ASR architecture.
>
> We agree that a paper discussing the establishment of the orthographic guidelines and other parts of this process would be beneficial for others and is something we want to do in the future.
>
> Ka tika, ka mihi anō ki a koe i te whai wā kia arotake i tēnei pepa. Once again, thank you for taking the time to review this paper.

---

> ### Comment · Reviewer_bjZD · 2024-08-26
> **Thank you to the authors for their reply**
>
> Thank you to the authors for the updated figures, which are clearer, and the additional detail they have provided on gender representation and ASR architecture. I continue to appreciate the attention given in this paper to language details that have cultural and symbolic importance to the language community. This paper, from my perspective, cogently and concisely makes the case for the importance of vocabulary, pronunciation and knowledge representations that accurately reflect the language community's usage, history and aspirations. Without respect for these aspects, language technology, dataset and benchmark developments for low resource languages cannot meet their ethical obligations to the communities that speak the languages. This is a very important perspective to have in this track. I maintain my original score.

---

### Official Review · Reviewer_Xhje · 2024-07-24
**interesting benchmark but need more details**

**Rating:** 5
**Confidence:** 4
**Correctness:** yes
**Clarity:** generally ok

**Review:**

The paper discusses the crucial issue of the lack of suitable benchmarks for low-resource and Indigenous languages. The authors commendably argue that popular benchmarks are ill-suited for these languages. The paper is well-structured and provides a detailed description of the limitations of existing benchmarks.
However, the dataset description and technical details of the ASR model are not sufficiently clear, and the figures in the analysis section are blurred and difficult to interpret. The discussion on benchmark limitations could be more specific, and the lack of public access to the dataset necessitates clearer descriptions for researchers.

**Strengths:**

- The motivation for focusing on low-resource and Indigenous languages is strong
- The paper provides a thorough critique of existing benchmarks, highlighting their general issues and specific shortcomings for low-resource languages
- The methodology for creating the Te Taumata benchmark is well-documented, emphasizing the involvement of the language community and the importance of cultural relevance
- The discussion on the importance of reflecting the diversity of speakers and the unique characteristics of the language is insightful

**Additional Feedback:**

none

**Documentation:**

generally ok, but more details of the model are needed

**Limitations:**

see above

**Opportunities For Improvement:**

- The title seems confusing to me. The authors emphasize that existing benchmarks do not meet the real needs of the language speakers. However, their discussion on the limitations are mainly about general benchmark issues (e.g., vocabulary range and annotation mistakes), which can occur in any dataset, including high-resource languages like English and Chinese. I would like to see more specific explanations on why these datasets do not *fit* the real use cases of low-resource and Indigenous languages.
- The description of the dataset and the technical details of the trained ASR model are not sufficiently clear. For example, providing a summary of the current status of the datasets and more detailed technical specifications would enhance the paper.
- The figures in the analysis section, particularly Figures 3 and 4, are blurred and difficult to interpret. Improving the clarity and resolution of these figures is necessary.

**Relation To Prior Work:**

yes

**Summary And Contributions:**

This paper presents a critique of existing AI benchmarks and proposes a new, culturally and linguistically appropriate benchmark for the Māori language. It highlights the inadequacies of popular benchmarks like FLEURS and FLoRes in serving the needs of low-resource and Indigenous languages and introduces Te Taumata, a benchmark tailored to the Māori language community. The paper underscores the importance of benchmarks that reflect the unique linguistic, cultural, and historical contexts of Indigenous languages.

---

> ### Author Rebuttal · Authors · 2024-08-16
>
> Thank you for reviewing our paper. Several of the issues you raised have been included in the global response and are left out of here as they were picked up by some or all reviewers. This rebuttal contains the remaining issues you raised:
>
> The technical details of the ASR model have been made clearer in section 3.1.
>
> We have reordered Section 2.1 to bring the cultural worldview discussion to the front and added a summary section to address how the existing benchmarks to not meet the real needs of the language speakers.  Sections 3.3 and 4 discuss how the benchmark we have created addresses those limitations and how the data selected better reflects real uses cases and better meets the purposes of the community.

---

### Official Review · Reviewer_f8XK · 2024-07-25
**Fit for our purpose, not yours: Benchmark for a low-resource, Indigenous language**

**Rating:** 6
**Confidence:** 4
**Correctness:** Yes
**Clarity:** Yes

**Review:**

To address the issue of inadequate benchmarks for the Maori language, the authors created their own benchmark using annotated data from audio transcriptions spoken by native Maori speakers. They compare the performance of their dataset and benchmark against established benchmarks such as FloRes and Fleurs, finding that their benchmark outperforms these in terms of diversity of utterances, number of speakers, distribution of sentences, and number of unique words.
This research highlights the importance of creating more accurate and culturally sensitive benchmarks for low-resource languages, ensuring that these languages are properly represented and preserved in the ever-evolving field of language modeling.

**Strengths:**

The transcription of audio by Native Maori speakers makes the dataset created by the authors particularly robust. While there is a note that some transcriptions may be less accurate due to an inability to decipher the words said with confidence, the overall inclusion of this dataset is a positive for scholarship in the space. The authors do a good job stating the importance of being community aware and focused when creating datasets of low-resource languages without falling into the trap of creating these datasets as a machine translation tool that prioritizes English.

The benchmark the authors have created is very effective at judging the true quality of Maori language output and will certainly be a welcome addition to research looking to preserve or properly represent the Maori language in the LLM space.

**Additional Feedback:**

NA

**Documentation:**

I was not able to find the code for this paper.

**Ethics:**

Perhaps, I am not clear on how much permission the authors received to transcribe audio of human subjects.

**Opportunities For Improvement:**

The authors would have gotten a higher rating if they would have spoken directly to the limitations and ethical concerns around creating a Maori language dataset and benchmark.

**Relation To Prior Work:**

Yes

**Summary And Contributions:**

The authors make the convincing case that outstanding benchmarks evaluating the performance of low resource languages are inefficient due to their reliance on machine translation models that are largely trained on English and underlying dataset that comes from sentences translated from Wikipedia into 101 languages. Unfortunately, these translations have frequent grammatical errors on the Maori language meaning performance on this benchmark is not an accurate reflection on the effectiveness of the model's training on said language. Because of this, the authors create their own benchmark using annotated data from audio transcriptions spoken by Maori speakers. The paper then compares the performance of their dataset and benchmark against FloRes and Fleurs benchmarks finding that it outperforms these benchmarks across diversity of utterances, number of speakers, distribution of sentences and number of unique words.

---

> ### Author Rebuttal · Authors · 2024-08-16
>
> Thank you for reviewing our paper. Two of the issues you raised have been included in the global response and are left out of here as they were picked up by some or all reviewers. This rebuttal contains the remaining issue you raised:
>
> In an attempt to anonymise the paper it wasn’t clear that the audio that was transcribed is under the guardianship of the ‘authors’. The wording has been changed to resolve this. Furthermore, in reviewing the paper we haven’t made it explicit that the benchmark dataset is made up of the archives and contemporary recordings we discuss in lines 194-207. Wording has been changed throughout the new document to rectify this.

---

### Official Review · Reviewer_chc9 · 2024-08-02
**Benchmark for speech&language research - Maori**

**Rating:** 4
**Confidence:** 4

**Review:**

In general, the paper is relatively hard to follow, especially in terms of the writing, pronouns used, and structure. The following highlights some key issues:
- The originality of the paper is unclear. For instance, in lines 174-177, the authors mentioned a previous dataset for the language. What is the different between that and the proposed dataset?
- The comparison with datasets such as FLoRes is only qualitative, a quantitative would be useful to statistically prove the problem in previous datasets.
- Comparison on amount of efforts needed to curate the dataset should be added for future work to consider to follow.
- The quality of the labelling process should be discussed, such as agreement level between annotators.
- The texts on all figures are not intelligible.
- No discussion on limitations and potential negative societal impact are provided.
- A conclusion paragraph would be useful to summarise the paper and ease of understanding.
- It requires major revisions to enhance the readability and potential impact of the paper.

**Strengths:**

- The paper works on an important problem of speech&language research for low-resource languages.
- Qualitative discussions on previous datasets are provided.
- The pipeline for the creation of a dataset for a low-resource language is detailed.

**Additional Feedback:**

Please see above.

**Clarity:**

Parts of the paper are hard to follow. A conclusion section would be beneficial.

**Correctness:**

The construction of the dataset is reasonable. Comprehensive quantitative analyses of issues in previous datasets are recommended to further convince the readers.

**Documentation:**

No access to the dataset is provided.

**Limitations:**

No discussion on limitations and potential negative societal impact are provided.

**Opportunities For Improvement:**

- The novelty and insight of the paper are limited; thus, it is important for authors to enhance their paper with the community's interests and potential impacts in mind.
- The writing of the paper needs to be revised. The texts on all figures are not intelligible. Conclusion and Limitations paragraphs should be added for better readability.
- More comparisons and experiments on the novelty of the data creation process, the dataset, and the labelling process are needed. Please see the details above.

**Relation To Prior Work:**

It is not clear how this work is different than other Maori-centric dataset, such as the one mentioned in lines 174-177.

**Summary And Contributions:**

The paper discusses qualitative problems in previous datasets (grammatical errors, poor pronunciation, limited vocabulary) for the Maori language. The authors introduce a curated benchmark dataset for the Maori language with more numbers of speakers, sentences and words.

---

> ### Author Rebuttal · Authors · 2024-08-16
>
> Thank you for reviewing our paper. Several of the issues you raised have been included in the global response and are left out of here as they were picked up by some or all reviewers. This rebuttal contains the remaining issues you raised:
>
> In an attempt to anonymise the paper, we may have inadvertently confused you. The benchmark dataset mentioned in lines 174-177 is in fact Te Taumata, the authors’ benchmark. The wording has been changed to resolve this issue (see lines 158-161 of new doc).
>
> We provide a summary of key statistics in Table 2, highlighting the difference in lexical diversity between the datasets.
>
> In terms of a contextual analysis, this is something that we have expanded on. This includes an analysis on the 100 most frequent bigrams (Figure 2), highlighting some key differences with the two datasets. For example, the Fleurs dataset shows characteristics of written text, whereas the proposed dataset is related to general conversation and storytelling. Furthermore the Fleurs dataset contains little to no relevance to Māori culture and world view, as seen in the top 10 bigrams.
>
> We will include in the Limitations section a statement that reads: “To achieve a high quality benchmark dataset that takes into careful consideration, there is significantly more effort required than utilising existing benchmarks that do not.”
>
> The process of how to do the above is outlined in lines 202-219 of the new document.
>
> Further evidence for the effort required can be seen in Table 2. With 4x the number of sentences, nearly 3x the number of words and 3x the number of unique words that were all curated, labeled and reviewed by at least two language experts.
>
> The labelling processed is outlined in lines 230-250 (202-219 of the new document). At lines 245-246 we state “When the audio was tough to decipher, the pair would discuss the possible options.” We can add that “and agree on the final label.” (lines 215-218 of new document)
>
> The paper identifies the negative societal impacts of existing benchmarks in lines 86-96. The overall aim of the paper is to demonstrate an approach that reduces these negative impacts with culturally relevant and appropriate methods. There are no potential negative societal impacts identified in this approach.
>
> The benchmark presented is the first of its kind for the Māori language. Lines 195-196 highlight that the drivers for the overall project have been the community. Perhaps the use of third person in an attempt to anonymise the paper is confusing. The wording has been changed to resolve this issue (see lines 175-176 of new document).

---

### Author Rebuttal · Authors · 2024-08-16

Firstly, thank you for reviewing our paper. We have found the feedback very useful in improving our paper. There are several sections we have added to the paper to address comments made by some or all reviewers.

We have added a Limitations section to the paper. It discusses the limited access to data necessary for developing similar, high quality benchmarks and how to appropriately gain access. This includes specific types of data (i.e bilingual), speakers (i.e female and children), level of fluency, and domains (i.e informal).

Another limitation we have addressed with the benchmark is that the data is predominantly from the regional variation of the language found in the Far North. Future work will address this with sub-benchmarks from other regional variations.

We have included some discussion of ethical considerations we encountered when creating the benchmark, including data sovereignty (i.e access, permissions, guardianship) throughout the paper. It is beyond the scope of the paper to go into too much detail on these subjects, however, we have included references to our other works that give more discussion on this topic.

We have added a conclusion paragraph to summarise the paper.

The updated images have been attached as a PDF. They have been made smaller to fit on one page, but on the updated version we have created they are larger.

We have prepared an updated version of the paper with the new sections and any issues identified in the individual reviews. However, in order to respond to the feedback, this has resulted in a figure, the limitations section and conclusion sitting on a 10th page. We are happy to send the paper through if required.

---

### Decision · Program_Chairs · 2024-09-26

**Decision:**

Accept (Poster)

**Comment:**

This is a very interesting paper that can influence future work in constructing benchmark datasets for low-resource languages. The influence may be small in the size of the community, as most people tend to work on the major languages (English, Chinese, etc), but the small community of researchers working on the low-resource languages would very much relate to and learn from the issues discussed in this paper.

This is a pretty unique paper, different from the mainstream NeurIPS papers, even within the D&B track. There is less focus on the quantitative aspects of the dataset, and much more focus on the process and detailed considerations in working with the community to build the benchmark dataset of Maori language. I especially appreciate the careful consideration in choosing the speakers, both the native (older) speakers, as well as the L2 speakers, younger and probably more likely to be users of language technologies built from this and similar resources.

The uniqueness of the paper probably led to the lukewarm reviews of some of the reviewers, but they all gave good comments to make this paper be more engaging for the typical NeurIPS researcher, and it looks like the authors will take these comments and incorporate them into the final camera-ready.

I personally learned a great deal from reading this paper, and I look forward to seeing it at the conference. I think everyone can learn from this paper, even if working on English, Chinese, etc, as research on language technology should be so much more and deeper than measuring LLMs' performance on MMLU (with all due respect to the researchers of those models and benchmarks).